# Genetic Mechanism for Antioxidant Activity of Endogenous Enzymes under Salinity and Temperature Stress in Turbot (*Scophthalmus maximus*)

**DOI:** 10.3390/antiox11102062

**Published:** 2022-10-19

**Authors:** Xinan Wang, Aijun Ma, Zhihui Huang, Zhibin Sun, Zhifeng Liu

**Affiliations:** 1Yellow Sea Fisheries Research Institute, Chinese Academy of Fishery Sciences, Qingdao 266071, China; 2Laboratory for Marine Biology and Biotechnology, Pilot National Laboratory for Marine Science and Technology, Qingdao 266071, China

**Keywords:** *Scophthalmus maximus*, antioxidant enzymes, salinity, temperature, genotype, interactions

## Abstract

Three antioxidant properties (corresponding to the enzymes superoxide dismutase (SOD), catalase, and glutathione peroxidase) were measured in the livers of *Scophthalmus maximus* under different salinities/temperatures (5, 10, 20, 30, and 40‰/17, 20, 23, 25, and 28 °C). Split-plot (SP) analysis, additive main effects, and multiplicative interaction (AMMI) and genotype × environment interaction (GGE) biplots were used to analyze genotype × salinity/temperature interactions for antioxidant properties. The results of the SP analysis show that the activity of the three antioxidant properties was significantly (*p* < 0.01) affected by salinity/temperature, antioxidant, and salinity/temperature × antioxidant interaction. The results of the AMMI analysis for salinity reveal that the effect of genotype, salinity, and genotype × salinity interaction on antioxidant properties reached a significant level (*p* < 0.001); 92.1065%, 2.6256%, and 4.4360% of the total sum of squares for antioxidant property activity were attributable to the effects of genotype, salinity, and genotype × salinity interaction, respectively. The results of GGE biplot analysis for salinity reveal differences in the activity ranking of the three antioxidant properties under five salinities; this difference expands with an decrease or increase in salinity from 30‰ (optimum salinity). A salinity of 5‰ had the strongest ability to identify the three antioxidant properties. The five experimental salinities were divided into one region, and SOD activity was the highest in this region. In a comprehensive analysis of stability and activity, SOD had the best activity and stability. The results of AMMI analysis for temperature reveal that genotype, temperature, and genotype × temperature interaction had significant effects on the antioxidant properties (*p* < 0.001); 82.4720%, 4.0666%, and 12.0968% of the total sum of squares for antioxidant property activity were attributable to the effects of genotype, temperature, and genotype × temperature interaction, respectively. The results of GGE biplot analysis for temperature reveal a large difference in the activity ranking of antioxidant properties between 17 °C and the other four temperatures, while only small differences in the activity rankings were detected among the other four temperatures. The difference in the activity ranking of antioxidant properties was greatest between the temperatures of 17 and 20 °C. A temperature of 17 °C showed the strongest ability to distinguish the three antioxidant properties. Additionally, the five test temperatures were grouped into one region, and comprehensive analysis of activity and stability showed that SOD had the best activity and stability.

## 1. Introduction

Temperature and salinity are important environmental properties that affect the physiological activities and antioxidant defense systems of fish in aquatic environments; accordingly, they have an important impact on the survival and growth of fish [1,2,3,4]. When the environmental salinity or temperature changes abnormally, it causes a variety of physiological stress reactions in fish, and the production of reactive oxygen species (ROS) increases accordingly [3,4,5]. ROS not cleared effectively damage the body of the fish, causing adverse effects such as enzyme inactivation, lipid peroxidation, and DNA damage [6,7], and also attack their immune and epidemic prevention mechanisms, further damaging the normal growth and survival of the fish. Fish have an antioxidant enzyme system that includes superoxide dismutase (SOD), catalase (CAT), and glutathione peroxidase (GPX) [8]. The antioxidant enzyme system plays an important role in removing excessive reactive oxygen free radicals, thereby alleviating self-damage and improving the defensive ability of immune cells [9]. In an environment with unsuitable salinity or temperature, the antioxidant enzyme system is an important immune defense used by fish to resist environmental stress [10].

The main method of culturing turbot in China is the running water industrialized culture mode of “greenhouse + deep well seawater”, which is inevitably affected by the salinity of groundwater. Although turbot has a wide range of tolerance to salinity [11], long-term inappropriate salinity will have a great impact on the growth of turbot, which hinders development. In addition, turbot (*Scophthalmus maximus*) is a cold water fish, its maximum temperature for growth is 21–22 °C, and its optimum growth temperature is 14–17 °C. The survival rate and growth of turbot are negatively impacted at water temperatures > 23° C [12,13]. Obviously, turbot can tolerate only a narrow range of temperatures, which means that temperature has a profound impact on turbot culture and the rearing of young. The natural seawater temperature in northern China is high in summer. Turbot show slow growth and a low survival rate when cultured under high-temperature environments. Therefore, studying the salinity and temperature tolerance of turbot and breeding salt- and heat-tolerant varieties are of great significance to aquaculture and the economy.

Since changes in antioxidant enzyme activity reflect the physiological status of fish under different environmental conditions to a certain extent, they can be used as a physiological index to measure the degree to which fish are under external environmental stress [14]. Therefore, antioxidant properties can be used as a salt- and heat-tolerance index for turbot breeding. Studies on the effect of fish salinity and temperature on antioxidant enzyme activity have been reported previously [15,16,17,18,19]. Wang et al. (2005) showed that the activities of SOD and CAT in the blood of *Sebastes schlegelii* increased gradually with a decrease in seawater salinity [15]. Liu et al. (2013) studied the effects of different salinities on the activities of SOD, CAT, and GPX in the liver of *Trachinotus ovatus* at different temperatures. The results show that, at 18 and 21 °C, SOD activity was higher than that in the control group at 1, 3, 6, 12, and 24 h, while CAT activity was significantly higher than that in the control group at 24 h; at 29 °C, the activities of SOD and CAT at the 24 h sampling point were significantly higher than those in the control group; and at 32 °C, the activities of SOD and CAT at the five sampling time points were significantly lower than those in the control group [16]. Li et al. (2021) [17] studied the effects of different temperatures on the activity of SOD in black sea bass (*Centropristis striata*) and reported that SOD activity decreased first and then increased with increasing temperature on day 1 of the experiment, but increased gradually with increasing temperature on day 60 of the experiment. Liu et al. (2016) [18] studied the activities of antioxidant enzymes in the liver and intestine of *Larimichthys polyactis* at different water temperatures. They found that the activities of SOD in the liver and intestine and CAT in the liver increased first and then decreased with increasing water temperature, but the effect of temperature on CAT activity in the intestine was not significant. Guo et al. (2012) studied the effects of different salinities/temperatures on the activities of antioxidant enzymes in the body surface mucus, blood, gills, and liver of *S. maximus* under different salinities/temperatures. The results showed that salinity/temperature had a significant effect on the activities of SOD, CAT, and GPX: when the temperature was constant and the salinity changed, the activities of various enzymes in the liver, gill, serum, and mucus were lower at normal seawater salinity (30‰) and increased with decreasing or increasing salinity; when the salinity was constant and the temperature increased, the changes in enzyme activity in each tissue were not significant [19]. These studies provide basic data on salinity/temperature tolerance breeding and the disease control of fish.

However, in animal and plant breeding, first, the genetic mechanism of breeding indexes should be studied, and then the breeding plan should be formulated on this basis. The activity of antioxidant properties is different under different salinities/temperatures. This difference may be attributed not only to different salinities/temperatures, but also to the effects of genotype itself and the interaction between genotype and salinity/temperature. Therefore, genetic analysis of the interaction between antioxidant and salinity/temperature is of great significance for an in-depth understanding of the genetic roles of SOD, CAT, and GPX and for determining the tolerance salinity/temperature range of *S. maximus*. However, for this topic, research on *S. maximus* has not been reported.

In this study, split-plot (SP) analysis, additive main effects, and multiplicative interaction (AMMI) [20] and genotype main effects and genotype × environment interaction (GGE) biplots [21] were used to analyze genotype × salinity/temperature interactions for antioxidant properties in *S. maximus*. The purpose of this study was to analyze the genetic mechanism of antioxidant properties in *S. maximus* in depth so that a reference could be provided for formulating a breeding plan for salinity/temperature tolerance using antioxidant properties as breeding indexes.

## 2. Materials and Methods

### 2.1. Ethics Statement

All experimental treatments for artificially cultivated fish were performed according to the recommendations in the Guide for the Care and Use of Laboratory Animals of the National Institutes of Health, China. The study protocol followed the recommendations of the Experimental Animal Ethics Committee, Yellow Sea Fisheries Research Institute, Chinese Academy of Fishery Sciences, China (decision no: YSFRI-2022021).

### 2.2. Experimental Materials

The young turbots were raised in Tianyuan Aquatic Limited Corporation in Yantai, China. The individual weight was 80.6 ± 6.3 g. They were all healthy young fish that were artificially bred. Five salinity/temperature levels were set in the experiment (5, 10, 20, 30 and 40‰/17, 20, 23, 25, and 28 °C) and three repetitions were set for each salinity/temperature level. The experiment was carried out in glass tanks, each containing about 300 L of seawater and stocking 30 fish. The experimental fish were kept in a glass tank for a week before the experiment began. Throughout the temporary holding and experiment, the fish were fed bait at 9:00 every day, the tank was aerated continuously, and the water was changed once a day. For the salinity part of the experiment: The normal salinity of seawater is 30‰. Low-salinity seawater was prepared from underground seawater and freshwater, and high-salinity seawater was prepared from underground seawater and sea crystal. The static water method was used to aerate the culture and an automatic heater was used to obtain a constant temperature of 17 °C. For temperature part of the experiment: The water temperature was increased to 17 °C from normal temperature (13 °C) for 24 h and then to each test temperature at a rate of 1 °C every 16 h. Static water aeration culture was used, and the temperature was controlled by automatic thermostatic heaters. The salinity was controlled at 30‰. In addition, under these two experimental conditions, the illumination intensity was 500−1500 lx, the pH was 7.6–8.2, the dissolved oxygen content was 7.8 ± 0.2 mg/L, and the TAN content was <0.1 mg/L. After staying at the set salinity/temperature for 48 h, three fish were randomly taken from each tank and anesthetized with MS-222 (tricaine methane sulfonate) (Maya Reagent Limited Corporation, Jiaxing, China). After disinfection, the liver of each fish was removed using scissors, rinsed gently with precooled distilled water, dried with filter paper, and placed in a 1.5 mL centrifuge tube. The samples were temporarily stored in a freezer at −20 °C for 24 h, and then transferred to a freezer at −80°C for use in the next part of the experiment.

### 2.3. Experimental Methods

#### 2.3.1. Preparation of Enzyme Solution

To prepare the liver samples for enzyme analysis, we removed each sample from the freezer, moistened it with distilled water at 4 °C, dried it with filter paper, added nine times the volume of normal saline (0.86%) per gram of tissue, and homogenized the mixture in an ice bath using a glass homogenizer. The homogenate was centrifuged at 12,000 rpm at 4 °C for 30 min, and the supernatant was removed to measure enzyme activities.

#### 2.3.2. Enzyme Activity Measurements

The activities of SOD, CAT, and GPX were measured using kits purchased from Nanjing Jiancheng Biological Engineering Research Institute (Jiangsu, China).

##### SOD

SOD activity was measured using the WST-1 method. The absorbance value was measured by colorimetry at 550 nm to calculate its activity. The unit of activity (U/mg) was defined as the amount of SOD corresponding to the inhibition rate of SOD per milligram of tissue protein in a 1 mL reaction solution that reached 50%.

##### CAT

CAT activity was measured using the visible light method, and the amount of H_2_O_2_ reduction was measured at 450 nm. The amount of H_2_O_2_ decomposed by 1 μmol per milligram of tissue protein per second was defined as one unit of activity (U/mg).

##### GPX

GPX activity was calculated by measuring the consumption of glutathione at 412 nm. The definition of one unit of activity was as follows: for every mg of protein, the effect of non-enzymatic reaction was deducted every minute to reduce the concentration of glutathione in the reaction system. The unit of enzyme activity was defined as the concentration of glutathione (GSH) in the reaction system reduced by 1 μmol/L (U/mg).

### 2.4. Data Analysis

#### 2.4.1. SP Analysis

This experiment was laid out as a split-plot design; salinity/temperature was set as the main-plot property, with the five salinity/temperature gradients (5, 10, 20, 30, and 40‰/17, 20, 23, 25, and 28 °C) assigned to five main plots in each of three complete replicate blocks, while antioxidant enzyme was set as the sub-plot property, with the three antioxidant enzymes (CAT, SOD, and GPX) assigned to three sub-plots within each main plot. The split-plot analysis model can be written as follows:(1)$$y_{mnp}=\mu +a_{p}+g_{mn}+h_{mp}+\varepsilon_{mp}$$
where $y_{mnp}$ is the activity of the *m*-th salinity/temperature treatment for the *n*-th antioxidant enzyme in the *p*-th complete block; $g_{mn}$, μ, and $a_{p}$ are the *mn*-th treatment effect, the general intercept, and the effect of the *p*-th block, respectively; $h_{mp}$ is the main-plot error associated with the *p*-th block and *m*-th activity gradient, assumed to be random with a mean of zero and a variance of $\sigma_{h}^{2}$; and $\varepsilon_{mp}$ is a residual sub-plot error with a mean of zero and a variance of $\sigma^{2}$.

#### 2.4.2. AMMI Analysis

The main feature of the AMMI model integrates the analysis of variance and principal component analysis [20]. The AMMI model for the *g*-th genotype (SOD, CAT, and GPX) in the *e*-th salinity/temperature (5, 10, 20, 30, and 40‰/17, 20, 23, 25, and 28 °C) can be written as follows:(2)$$y_{ge}=\mu +\rho_{g}+\psi_{e}+\overset{M}{\underset{i=1}{\sum }}\theta_{n}\xi_{gn}\eta_{en}+\varepsilon_{ge}$$
where $y_{ge}$ is the activity of antioxidant enzymes for genotype *g* at salinity/temperature *e*; $\psi_{e}$, μ, and $\rho_{g}$ are the average deviation of the salinity/temperature, the grand mean, and the average deviation of genotypes, respectively; $\eta_{en}$ is the salinity/temperature principal component score of the *n*-th principal component; $\xi_{gn}$ is the genotype principal component score of the *n*-th principal component; $\theta_{n}$ is the eigenvalue of the *n*-th interaction principal component axis; *M* is the total number of principal component axes; and $\varepsilon_{ge}$ is the residual.

#### 2.4.3. GGE Biplot Analysis

GGE biplot analysis can reveal the complex interaction between different properties [22,23,24]. The antioxidant activity data obtained from fish cultured at different salinities/temperatures were sorted into a two-way table that included salinity/temperature and antioxidant enzyme activity. For each mean value of activity, there was a corresponding antioxidant property at the corresponding salinity/temperature. Singular value decomposition of the first two principal components was used to fit the GGE biplot model [21] as follows:(3)$$y_{ge}=\mu +\psi_{e}+\theta_{1}\xi_{g1}\eta_{e1}+\theta_{2}\xi_{g2}\eta_{e2}+\varepsilon_{ge}$$
where $y_{ge}$ is the trait mean activity for genotype *g* at salinity/temperature *e*; μ, $\psi_{e}$, and $\mu +\psi_{e}$ are the grand mean, the main effect of salinity/temperature *e*, and the mean activity across all genotypes at salinity/temperature *e*, respectively; $\eta_{e1}$ and $\eta_{e2}$ are eigenvectors of salinity/temperature *e* for PC1 and PC2, respectively; $\xi_{g1}$ and $\xi_{g2}$ are eigenvectors for genotype *g* for PC1 and PC2, respectively; $\theta_{1}$ and $\theta_{2}$ are the singular values for the first and second principal components (PC1 and PC2), respectively; and $\varepsilon_{ge}$ is the residual associated with genotype *g* at salinity/temperature *e*. 

The SP, AMMI, and GGE biplot analyses were conducted using the DPS data processing system [25].

## 3. Results

### 3.1. SP Analysis of Variance

#### 3.1.1. SP Analysis of Variance for Salinity Resistance Experiment

The results of the SP analysis of variance for the salinity resistance experiment are listed in Table 1. Table 1 shows that the *p* values for salinity, antioxidant, and salinity × antioxidant interaction are 0.0002, 0, and 0, respectively, indicating that the activity of the three antioxidant properties was significantly (*p* < 0.01) affected by salinity, antioxidant, and salinity × antioxidant interaction.

#### 3.1.2. SP Analysis of Variance for Temperature Resistance Experiment

The results of the SP analysis of variance for the temperature resistance experiment are listed in Table 2. Table 2 shows that the *p* values of temperature, antioxidant, and temperature × antioxidant interaction are 0.0002, 0, and 0, respectively, indicating that the activity of the three antioxidant properties was significantly (*p* < 0.01) affected by temperature, antioxidant, and temperature × antioxidant interaction.

### 3.2. AMMI Analysis of Variance

#### 3.2.1. AMMI Analysis Results for Different Salinity Trials

The results of AMMI analysis for different salinity trials show that antioxidant property activities were significantly affected by genotype, salinity, and genotype × salinity interaction. Overall, 92.1065% of the total sum of squares (SS) for antioxidant property activity was captured by the effect of genotype, while and 2.6256% and 4.4360% of the total SS were attributable to the effects of salinity and genotype × salinity interaction, respectively. IPCA1 was obtained, which contributed to 99.9429% of the effect of genotype × salinity interaction (Table 3).

#### 3.2.2. AMMI Analysis Results for Different Temperature Trials

The results of AMMI analysis for different temperatures trials show that the activities of antioxidant properties were significantly affected by genotype, temperature, and genotype × temperature interaction (Table 4). The AMMI analysis of variance indicated that 82.4720% of the total sum of squares (SS) for antioxidant property activity was captured by the effect of genotype, and that 4.0666% and 12.0968% of the total SS were attributable to the effects of temperature and genotype × temperature interaction, respectively. IPCA1 was obtained, which contributed to 90.11606% of the genotype × temperature interaction.

### 3.3. GGE Biplot Analysis

GGE biplot analysis was carried out based on the mean activities of three antioxidant properties at five salinities/temperatures. The “relationship among different salinities/temperatures” (Figure 1A and Figure 2A), “which-won-where” (Figure 1B and Figure 2B), “high activity and activity stability” (Figure 1C and Figure 2C), and “concentric circles” views of the GGE biplot (Figure 1D and Figure 2D) were based on the results from the GGE biplot analysis as shown in Table 5 and Table 6.

The GGE biplots of the relationship among different salinities/temperatures (Figure 1A and Figure 2A) mainly analyze the similarity of antioxidant properties among salinities/temperatures. The included angle of the two line segments indicates the correlation of the activity ranking of the measured antioxidant properties under the salinity/temperature represented by two line segments. When the included angle of the two segments is an acute angle, the activity ranking of antioxidant properties under the two salinities/temperatures has a positive correlation. The smaller the angle and the higher the correlation, the closer the activity ranking of antioxidant properties. When the angle between the two segments is an obtuse angle, the activity ranking of antioxidant properties is negatively correlated under the two salinities/temperatures. The “which-won-where” view of the GGE biplot (Figure 1B and Figure 2B) divides the experimental regions according to the interaction between antioxidant and salinity/temperature and reveals the antioxidant property with the highest activity level in each region. The antioxidant property located on the top corner of the polygon in each region is the antioxidant property with the highest activity in this region. The “high activity and activity stability” view of the GGE biplot (Figure 1C and Figure 2C) can determine the antioxidant property with high and stable activity. The direction of the transverse oblique line to the right represents the approximate average activities of antioxidant properties for all salinities/temperatures. The straight line perpendicular to the transverse slash represents the tendency of antioxidant × salinity/temperature interaction. The more deviated from the transverse oblique line, the more unstable the vertical line. The GGE biplot view with concentric circles (Figure 1D and Figure 2D) judges the high activity and activity stability based on the distance from various antioxidant properties to the central point of antioxidant properties. The smaller the distance, the higher and more stable the activities of antioxidant properties.

#### 3.3.1. GGE Biplot Analysis for Different Salinity Trials 

In Figure 1A, the included angles among the five salinities are acute angles, which indicates that the correlation of the activity ranking of the three antioxidant properties under the five salinities is positive. Among them, the included angle under the two salinities of 10‰ and 40‰ is the smallest (close to 0), indicating that the activity ranking of the three antioxidant properties is almost the same under the two salinities, whereas the included angle under the two salinities of 5‰ and 30‰ is the largest, indicating that the activity ranking difference under the two salinities is the largest. The length of the line segment represents the ability of the salinity to distinguish antioxidant properties. The longer the line segment, the stronger the ability to distinguish. In Figure 1A, 5‰ has the strongest ability to distinguish the three antioxidant properties, followed by 40‰, 20‰, 10‰, and 30‰. In Figure 1B, five test salinities are divided into one region, and SOD has the highest activity in this region. In Figure 1C, the average activity of SOD is the highest, followed by GPX and CAT; the most stable activity corresponds to SOD, followed by CAT and GPX. In Figure 1D, SOD has the best in activity and stability, followed by GPX and CAT.

#### 3.3.2. GGE Biplot Analysis for Different Temperature Trials 

Figure 2A shows that the included angles among the five temperatures are all acute angles, which indicates that the correlation of the activity rankings of the three antioxidant properties under the five temperatures is positive. Among them, the included angles between 20 and 25 °C and between 25 and 28 °C were the smallest, indicating that the activity rankings of the three antioxidant properties were more similar under these two groups of temperatures. The included angle between 17 and 20 °C was the largest, indicating that the activity ranking difference under these two temperatures was the greatest. The length of the line segment indicates the ability of the temperature to distinguish antioxidant properties, with a longer segment indicative of a stronger ability to distinguish them. In Figure 2A, 17 °C has the strongest ability to distinguish the three antioxidant properties, followed by 28, 25, 23, and 20 °C. Figure 2B shows that the five test temperatures are grouped into one region, and SOD has the highest activity in this region. In Figure 2C, the average activity of SOD is the highest, followed by GPX and CAT; CAT has the most stable activity, followed by SOD and GPX. In Figure 2D, SOD has the best activity and stability, followed by GPX and CAT.

## 4. Discussion

Salinity and temperature are important properties that induce oxidative stress in fish. Under salinity/temperature stress, fish accumulate a large amount of ROS, causing oxidative damage, and macromolecular substances such as lipids, nucleic acids, and sugars are oxidized, resulting in metabolic dysfunction in the fish [26]. Therefore, ROS must be cleared rapidly and effectively. SOD, CAT, and GPX play decisive roles in scavenging ROS free radicals produced in the process of oxidative stress [27]. SOD can react with free radicals to produce H_2_O_2_. CAT can reduce H_2_O_2_ into oxygen molecules and water, or H_2_O_2_ can be cleared by GPX [28], thereby preventing oxidative damage by free radicals to biological macromolecules in the body, and maintaining the normal physiological activities of cells and the body. Thus, changes in SOD, CAT, and GPX activities can reflect the physiological conditions of organisms under different stress conditions [18], and they can be used as an indirect selection index for breeding turbot with salinity/temperature resistance.

The liver is one of the most active metabolic organs in the body. It plays an important role in regulating osmotic pressure and maintaining normal physiological activities in fish. It can decompose metabolites and toxic substances in the body. The content of ROS in the liver is an important index for evaluating the oxidative stress response and antioxidant defense ability of fish [29]. Therefore, in the current study, we used turbot livers to study the activity of antioxidant enzymes.

Considering the significance of antioxidant enzymes in scavenging free radicals in fish, salt/heat tolerance breeding with antioxidant enzymes as an indirect selection index is a method that is worth exploring. Studies on the breeding of turbot using antioxidant properties as breeding indexes have been reported. Guo et al. (2012) studied the effects of different temperatures and salinities on antioxidant enzyme activities in juvenile turbot. The results showed that salinity and temperature had a significant effect on the activities of SOD, CAT, and GPX. The activities of SOD, CAT, and GPX were lower at a salinity of 30‰ and higher at a salinity of 5 and 40‰. Their activities increased with a decrease or increase in salinity from 30‰; the activities of SOD, CAT, and GPX fluctuated with an increase in temperature, but overall their activities changed significantly with increasing temperature [19]. These conclusions provide basic data for salt/heat tolerance breeding of turbot. However, the activity of antioxidant properties under different salinities/temperatures is affected not only by the effect of salinity/temperature, but also by genotype and the interaction between salinity/temperature and genotype. Therefore, it is of great significance to analyze the effect of the interaction between genotype and salinity/temperature with antioxidant properties and clarify the genetic mechanism of antioxidant properties for the formulation of salt- and heat-tolerant breeding in turbot.

In the current study, the interactions between antioxidant, genotype, and salinity/temperature in turbot were determined using SP, AMMI, and GGE biplot analyses. The results of the SP analysis showed that the activity of the three antioxidant properties was significantly (*p* < 0.01) affected by salinity/temperature, antioxidant, and salinity/temperature × antioxidant interaction, which indicates that it is worthwhile to investigate genotype × salinity/temperature interactions for *S. maximus* antioxidant properties.

The results from AMMI analysis for salinity revealed that the effects of genotype, salinity, and genotype × salinity interaction on antioxidant properties reached a significant level (*p* < 0.001) within the current experimental salinity range (5–40‰). The genotype effect accounted for 92.1065% of the total sum of squares for antioxidant activity, while the effect of salinity and genotype × salinity interaction accounted for only 7.0616% of the total sum of squares, which showed that the difference in antioxidant property activity was mainly due to the genotype effect. Genotypes effect play a decisive role in the activity of antioxidants, which may be related to the wide range of salt tolerance in turbot. Studies have shown that adult turbot can grow in a salinity range of 12–40, and even survive in a water environment with salinity as low as 5 [11,30]. Therefore, it is not ideal to use the current salinity range (5–40‰) for a salt tolerance test. To show the dominant effect of salinity in a salt tolerance test with turbot, it is necessary to further expand the upper and/or lower limits of salinity. The results from the GGE biplot analysis for salinity indicate differences in the activity ranking of the three antioxidant properties under five salinities, except between 10‰ and 40‰ salinity. These differences expand with a decrease or increase in salinity from 30‰ (optimum salinity), which indicates that the greater the difference between stressful salinity and optimum salinity, the greater the difference in the mechanism of the physiological and biochemical reaction of antioxidant properties. The activity rankings of the three antioxidant properties at salinities of 10‰ and 40‰ are almost the same, which shows that the physiological and biochemical reaction mechanism of antioxidant properties is the same under a low salt concentration of 10‰ and a high salt concentration of 40‰. A salinity of 5‰ had the strongest ability to identify the three antioxidant properties. According to the interaction between antioxidant and salinity, the five experimental salinities were divided into one region, and SOD activity was the highest in this region. In a comprehensive analysis of stability and expression, SOD had the best activity and stability. In conclusion, the differences in the activities of antioxidant properties corresponding to SOD, CAT, and GPX under different salinities could be attributed to the different antioxidant response mechanisms under different levels of salinity stress. When the salinity changes, the antioxidant properties that play a leading role in the five salinities are different. 

The results from AMMI analysis for temperature reveal that genotype plays a decisive role in antioxidant activity, likely for the following two reasons: (1) even under extreme high-temperature conditions (28 °C), the activity of antioxidant properties still depends mainly on the genotype effect rather than on the effects of temperature and the interaction between temperature and genotype; (2) with increasing stressful temperature, the activity of antioxidant enzymes in turbot increases to deal with the free radicals, and the temperature effect is strong at first; however, when the temperature exceeds the tolerance limit, the body weakens, the enzyme activity decreases, and the temperature effect decreases. When the temperature effects of the two stages are combined, the temperature effect does not dominate the impact on antioxidant properties within the temperature range tested in this study. Turbot have strict requirements for temperature and other environmental indicators. The suitable water temperature for their growth is 14–17 °C; the maximum temperature at which they can grow is 21–22 °C; their upper temperature limit is 25–26 °C (though they cannot stay at this temperature for very long); and 28 °C is their lethal temperature [11,12]. In our study, the temperature effect on turbot was not dominant, and we believe the second reason described above may have been at work. Therefore, using the current temperature range of 17 to 28 °C to carry out high-temperature resistance tests was not ideal; the upper limit of the test temperature needs to be decreased. The GGE biplot analysis for temperature revealed some differences in the activity rankings of the three antioxidant properties under the five temperatures. There was a large difference in the activity rankings of antioxidant properties between 17 °C and the other four temperatures, while the difference among the other four temperatures was small. This is because 17 °C was the upper limit of suitable water temperature for turbot growth, whereas the other four temperatures were stressful temperatures for turbot growth. Obviously, the reaction mechanism of antioxidant properties under an appropriate water temperature was quite different from that under a high temperature, and the reaction mechanisms among the high temperature groups were more similar. The difference in the activity rankings of antioxidant properties between 17 °C and 20 °C was the largest because 20 °C is close to the growth limit temperature of turbot. When the growth limit temperature is exceeded, the body becomes weak and the enzyme activities decrease. A temperature of 17 °C had the strongest ability to distinguish the three antioxidant properties, which showed that the difference in the reaction mechanism among the three antioxidant properties was greatest at the appropriate water temperature for growth. The other GGE biplots show that the five test temperatures can be grouped into one region, and that SOD had the best activity and stability. In conclusion, the observed differences in the activities of SOD, CAT, and GPX under different stressful temperatures were likely due to the different antioxidant response mechanisms at work under different culture temperatures, and the antioxidant properties that played a leading role in the response to temperature change differed at different temperatures.

## 5. Conclusions

Salinity/temperature and salinity/temperature × antioxidant interaction have a significant effect on the activity of *S. maximus* antioxidant properties. Despite antioxidant properties reaching a significant level in response to salinity/temperature, the salinity/temperature accounted for only 2.6256%/4.0666% of the total sum of squares. Therefore, salinity/temperature was not the major contributor to the total variation in antioxidant enzyme activity, which indicates that using the current salinity/temperature range (5–40‰/17–28 °C) to carry out salinity and temperature resistance tests is not ideal. The upper and/or lower limits of salinity need to be expanded further, and the upper limit of the test temperature needs to be decreased. In a comprehensive analysis of activity and stability, SOD has the best activity and stability for both salinity and temperature, which indicates that, within the current experimental salinity/temperature range, SOD should be the preferred breeding index for the salinity/temperature-resistant breeding of turbot. Generally, primary and secondary lipid oxidation compounds (LPO and MDA) are also used as important indicators of lipid peroxidation. However, the purpose of this study was to screen breeding indexes for salinity/temperature tolerance breeding based on the genetic mechanism of antioxidant properties. MDA content is the final reaction of the interaction between oxygen free radicals and the redox system in the body under external stress, and it cannot purely reflect the degree of external stress or the antioxidant capacity, but rather the interaction between the two; at the same time, LPO is unstable. Therefore, LPO and MDA were not measured in this study.

## Figures and Tables

**Figure 1 antioxidants-11-02062-f001:**
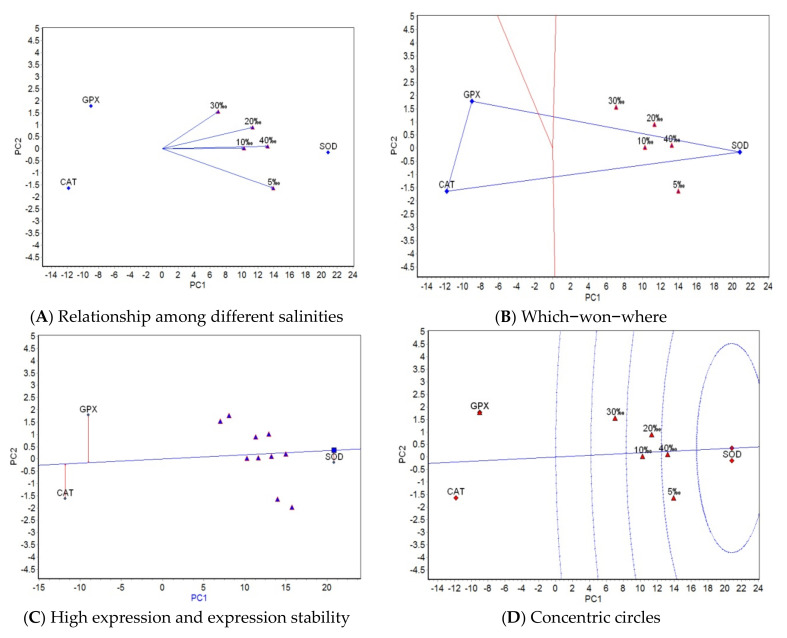
GGE biplots for antioxidant properties in different salinities.

**Figure 2 antioxidants-11-02062-f002:**
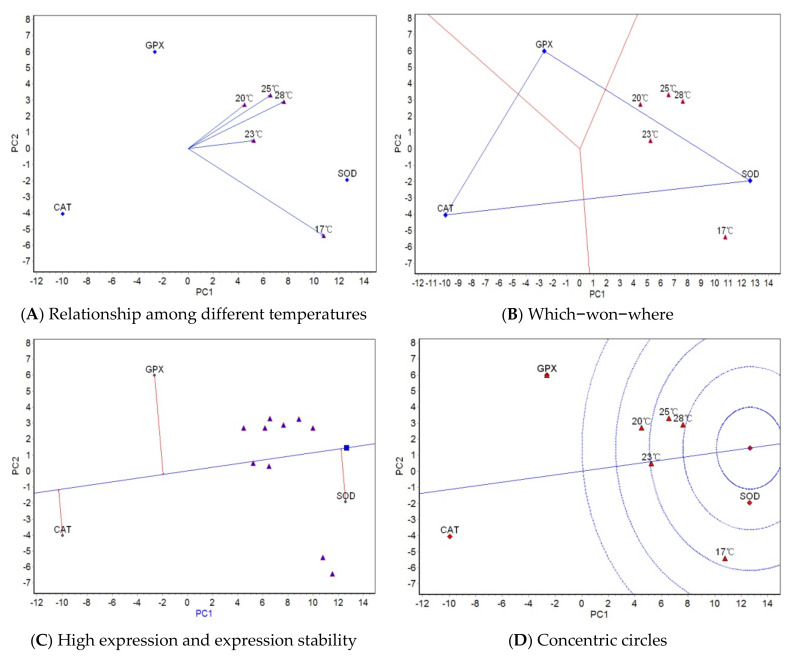
GGE biplots for antioxidant properties in different temperatures.

**Table 1 antioxidants-11-02062-t001:** Split-plot analysis of variance for *S. maximus* salinity resistance experiment, with five salinity gradients and three antioxidant enzymes.

Scheme	Sum of Squares	Degrees of Freedom	Mean Square	*F*-Value	*p*-Value
Blocks (replicates)	626.4706	2	313.2353		
Salinity	34,937.229	4	8734.3073	21.5390 **	0.0002
Main-plot error	3244.0959	8	405.5120		
Antioxidant	1,225,619.3	2	612,809.6700	1702.3200 **	0
Salinity × antioxidant	59,027.97	8	7378.4963	20.4970 **	0
Split-plot error	7199.7012	20	359.9851		
Total	1,330,654.8	44			

Notes: Asterisks denote that correlations were significant at ** *p* < 0.01.

**Table 2 antioxidants-11-02062-t002:** Split-plot analysis of variance for *S. maximus* temperature resistance experiment, with five temperature gradients and three antioxidant enzymes.

Scheme	Sum of Square	Degrees of Freedom	Mean Square	*p*-Value	*p*-Value
Blocks (replicates)	223.7234	2	111.8617		
Temperature	9480.4305	4	2370.1076	23.332 **	0.0002
Main-plot error	812.6712	8	101.5839		
Antioxidant	192,265.89	2	96,132.944	896.469 **	0
Temperature × antioxidant	28,201.121	8	3525.1401	32.873 **	0
Split-plot error	2144.7023	20	107.2351		
Total	233,128.54	44			

Notes: Asterisks denote that correlations were significant at ** *p* < 0.01.

**Table 3 antioxidants-11-02062-t003:** AMMI analysis of antioxidant enzymes for different salinity trials.

Source of Variation	*df*	SS	MS	*P*	Prob.	% of Total SS
Total	44	1,330,655	30,242.15			
Treatment	14	1,319,585	94,256.04	255.4302 **	0	
Genotype	2	1,225,619	612,809.7	1660.691 **	0	92.1065
Salinity	4	34,937.23	8734.307	23.6696 **	0	2.6256
Interaction	8	59,027.97	7378.496	19.9954 **	0	4.4360
IPCA1	5	58,994.23	11,798.85	31.97442 **	0	99.9429
Residual	3	33.73659	11.24553			
Error	30	11,070.27	369.0089			

Notes: *df*—degree of freedom; SS—sum of squares; MS—mean of squares. ** Significant at the 1% probability level.

**Table 4 antioxidants-11-02062-t004:** AMMI analysis results of antioxidant enzyme activities for different temperatures trials.

Source of Variation	*df*	SS	MS	*P*	Prob.	% of Total SS
Total	44	233,128.5	5298.376			
Treatment	14	229,947.4	16,424.82	154.8977	0	
Genotype	2	192,265.9	96,132.94	906.6019 **	0	82.4720
Temperature	4	9480.431	2370.108	22.3518 **	0	4.0666
Interaction	8	28,201.12	3525.14	33.2446 **	0	12.0968
IPCA1	5	25,413.74	5082.748	47.93392 **	0	90.1160
Residual	3	2787.381	929.1271			
Error	30	3181.097	106.0366			

Notes: *df*—degree of freedom; SS—sum of squares; MS—mean of squares. ** Significant at the 1% probability level.

**Table 5 antioxidants-11-02062-t005:** GGE biplot analysis results for antioxidant enzymes in different salinity trials.

Antioxidant Enzyme/ Salinity	Mean Activity	Deviation	PCA1	PCA2	PCA3	Distance from Center Point (D_i_)
SOD	366.434	232.666	20.824	−0.151	1.143 × 10^−7^	20.824
CAT	1.59	−132.178	−11.801	−1.627	1.143 × 10^−7^	11.912
GPX	33.28	−100.488	−9.023	1.779	1.143 × 10^−7^	9.196
5‰	164.083	30.315	13.980	−1.629	4.907 × 10^−8^	14.074
10‰	122.733	−11.035	10.269	0.032	−7.639 × 10^−8^	10.269
20‰	136.917	3.149	11.343	0.891	1.413 × 10^−7^	11.378
30‰	86.64	−47.128	7.0443	1.541	−2.075 × 10^−8^	7.2109
40‰	158.467	24.699	13.234	0.113	−1.027 × 10^−7^	13.234

**Table 6 antioxidants-11-02062-t006:** GGE biplot analysis results for antioxidant enzyme activities in different temperature trials.

Antioxidant Enzyme/ Temperature	Mean Activity	Deviation	PCA1	PCA2	PCA3	Distance from Center Point (D_i_)
SOD	159.718	85.9907	12.60829	−1.94039	0	12.75672
CAT	1.354	−72.3733	−9.96583	−4.04081	0	10.75388
GPX	60.11	−13.6173	−2.64246	5.981191	0	6.538901
17 °C	86.64	12.9127	10.7845	−5.40401	−1.8 × 10^−17^	12.0627
20 °C	56.8767	−16.8507	4.492894	2.702741	−1.8 × 10^−17^	5.243177
23 °C	55.6267	−18.1007	5.224673	0.479722	9.92 × 10^−17^	5.24665
25 °C	80.7333	7.006	6.572334	3.287603	−7.7× 10^−18^	7.348735
28 °C	88.76	15.0327	7.634451	2.884662	−2.5× 10^−17^	8.161257

## Data Availability

The data sets analyzed during the current study are available from Appendix A.

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
