# Peer review of "Genetic Mechanism for Antioxidant Activity of Endogenous Enzymes under Salinity and Temperature Stress in Turbot (Scophthalmus maximus)"

_antioxidants, 2022, doi:10.3390/antiox11102062_

Round 1

Reviewer 1 Report

The manuscript includes an interesting study. A high mathematical analysis was developed on the basis of a short number of biochemical determinations. The discussion is highly mathematical, and I find several aspects connected with biochemistry have not been addressed correctly. Concrete comments would be as follows:

Title

The authors have measured the activity of three antioxidant endogenous enzymes. Lipid oxidation development has not been measured. I would recommend modifying the title: “Genetic … for antioxidant activity of endogenous enzymes under …”

In general, the word “factors” could be replaced by “properties” throughout the whole paper, or sometimes by the words “endogenous enzyme activity”. The term factor seems more mathematic than biochemical.

Abstract

Line 12: The living time under such conditions is not expressed. It is not clear what living conditions have been carried out exactly. Was an RSM analysis carried out, taking into account all such values for both variables ? This has to be clarified in this section.

Introduction

Line 42: affect.

This section seems too long. It could be shortened.

Materials and methods

Line 138: It is not clear what living conditions were carried out. Five levels are mentioned to be studied, but there are two variables, each of them having 5 different values. As mentioned above, was there an RSM analysis applied ?

Each living condition was developed for 48 h. Are the authors sure that this time would be enough for having an effect on the endogenous enzyme activity ? Did the authors try with longer periods ?

Samples were stored at -20 ºC. How long ? This temperature is relatively high; a lower temperature ought to have been employed. Activity of endogenous enzymes may be modified during such storage. This fact provides many doubts about the results obtained.

The number of replicates carried out is not expressed.

Results and discussion

Only activity of three endogenous enzymes was measured. Determination of primary and secondary lipid oxidation compounds would have been very interesting. This would have complemented the results obtained and would be a real determination of lipid oxidation development. No indication of the need of this kind of analyses is mentioned in the Conclusions section.

Author Response

Dear Reviewer:

We are very grateful for the comments from the reviewer. According to these good comments, we have revised the manuscript. The changes of the manuscript were highlighted by using “Track Changes”.

Title

  1. The authors have measured the activity of three antioxidant endogenous enzymes. Lipid oxidation development has not been measured. I would recommend modifying the title: “Genetic … for antioxidant activity of endogenous enzymes under …”.
    Response:

According to the comment of the reviewer, we changed the title to “Genetic mechanism for antioxidant activity of endogenous enzymes under salinity- and temperature- stress in turbot (Scophthalmus maximus)".
2. In general, the word “factors” could be replaced by “properties” throughout the whole paper, or sometimes by the words “endogenous enzyme activity”. The term factor seems more mathematic than biochemical.
Response:

According to the comment of the reviewer, we replaced the word “factors” with “properties” throughout the whole paper.

Abstract

  1. Line 12: The living time under such conditions is not expressed. It is not clear what living conditions have been carried out exactly. Was an RSM analysis carried out, taking into account all such values for both variables ? This has to be clarified in this section.

Response:

According to the comment of the reviewer, we added the living time in Line 12.

According to the comment of the reviewer, we added the  exact living conditions in “2.2. Experimental materials” section.

According to the comment of the reviewer, we tried to conducted RSM analysis. However, in this way, we found a problem. In the experimental design of this manuscript, salinity and temperature are two independent experiments respectively. The five salinity experiments was carried out at 17℃, and the five temperature experiments was carried out under salinity of 30‰. Therefore, this experiment is not a complete interaction of temperature and salinity. That is, only the interaction between 17℃ and five salinities, and the interaction between salinity 30‰ and five temperatures. Therefore, such an experimental design cannot be used for RSM analysis.

Introduction

  1. Line 42: affect.

Response:

According to the comment of the reviewer, we changed "affects" to "affect" in line 42.

  1. This section seems too long. It could be shortened.

Response:

According to the comment of the reviewer, we have shortened “Introduction” section.

Materials and methods

  1. Line 138: It is not clear what living conditions were carried out. Five levels are mentioned to be studied, but there are two variables, each of them having 5 different values. As mentioned above, was there an RSM analysis applied ?

Response:

According to the comment of the reviewer, According to the comment of the reviewer, we added the  exact living conditions in “2.2. Experimental materials” section in Line 138.

According to the comment of the reviewer, we tried to conducted RSM analysis. However, in this way, we found a problem. In the experimental design of this manuscript, salinity and temperature are two independent experiments respectively. The five salinity experiments was carried out at 17℃, and the five temperature experiments was carried out under salinity of 30‰. Therefore, this experiment is not a complete interaction of temperature and salinity. That is, only the interaction between 17℃ and five salinities, and the interaction between salinity 30‰ and five temperatures. Therefore, such an experimental design cannot be used for RSM analysis.

According to the comment of the reviewer, we will design a complete interaction experiment of temperature and salinity in the future. First, we will carry out the RSM analysis of temperature and salinity interactiont, and on this basis, the genotype × salinity/temperature interactions of antioxidant at different temperature/salinity stress will be analyzed.

  1. Each living condition was developed for 48 h. Are the authors sure that this time would be enough for having an effect on the endogenous enzyme activity? Did the authors try with longer periods?.

Response:

Is the duration of 48 hours enough for having an effect on the endogenous enzyme activity? The reviewer raised a very important scientific question for us. This needs to be answered by carrying out research on the genetic mechanism of salinity/temperature activity at different times. According to the comment of the reviewer, next, we will carry out this research.
3. Samples were stored at -20 ºC. How long? This temperature is relatively high; a lower temperature ought to have been employed. Activity of endogenous enzymes may be modified during such storage. This fact provides many doubts about the results obtained.
Response:

The temperature and salinity experiment was carried out in Tianyuan Aquatic Limited Corporation in Yantai, China. Due to the limitation of experimental conditions, we temporarily stored the samples in a freezer at - 20℃. The next day, we took the samples back to Yellow Sea Fisheries Research Institute, Chinese Academy of Fishery Sciences, Qingdao, China to store in a freezer at - 80 ℃. According to the comment of the reviewer, we revised the sample preservation conditions as follows: “The samples are temporarily stored in a freezer at - 20℃ for 24 hours, and then transferred to a freezer at - 80℃ for use in the next experiment.”.
4. The number of replicates carried out is not expressed.
Response:

According to the comment of the reviewer, we supplemented the number of replicates in "2.2 Experimental materials" section.

Results and discussion
1. Only activity of three endogenous enzymes was measured. Determination of primary and secondary lipid oxidation compounds would have been very interesting. This would have complemented the results obtained and would be a real determination of lipid oxidation development. No indication of the need of this kind of analyses is mentioned in the Conclusions section.

Response:

Generally, primary and secondary lipid oxidation compounds (LPO and MDA) are also used as important indicators of lipid peroxidation. However, the purpose of this study was to screen breeding indexess for salinity/temperature tolerance breeding based on the genetic mechanism of antioxidant properties. Considering that MDA content is the final reaction of the interaction between oxygen free radicals and the redox system in the body under external stress, and it cannot purely reflect the degree of external stress or the antioxidant capacity, but the interaction between the two; at the same time, LPO is unstable. Therefore, LPO and MDA were not measured in this study.

According to the comment of the reviewer, we supplemented the reason why primary and secondary lipid oxidation compounds were not determined in the " Conclusions" section.

Reviewer 2 Report

The authors have established that salinity/temperature and salinity/temperature x antioxidant factor interaction have significant effect on the activity of antioxidant factors in turbot. The weakness with the study however, is that the upper and lower limits of salinity should be expanded further in order to obtain significant effects. It is concluded that the salinities and temperatures used in the present study were not enough. The study would have benefited largely by including such values in the study.

I would encourage resubmission after inclusion of further studies with a wider range of salinities and temperatures. The paper is well written, the introduction is comprehensive, the cited papers are adequate, and presentation is clear. 

Author Response

Dear Reviewer:

We are very grateful for the comments from the reviewer. For the comments of the reviewer, our response is as follows:

In this manuscript, we have carried out dissection of genotype × salinity/temperature interactions for antioxidant factors, including superoxide dismutase (SOD), catalase and glutathione peroxidase, in S. maximus using SP analysis, AMMI and GGE Biplot analysis.

As the activity of antioxidant enzymes is affected not only by salinity/temperature, but also by PH, ammonia nitrogen and other environmental factors, the present research is carried out under specific environmental factors such as PH, ammonia nitrogen. When PH, ammonia nitrogen and other environmental factors change, the research conclusions in this manuscript will change. That is to say, in each different environment, the tolerance salinity/temperature range needs to be re determined. Especially when the environmental conditions such as PH and ammonia nitrogen are quite different.

Therefore,the most outstanding contribution of current research is to provide a method to determine a reasonable stress resistance experimental interval under specific environmental factors such as PH, ammonia nitrogen.

In addition,according to the comment of the reviewer, we tried to conducted further studies with a wider range of salinities and temperatures.  However, considering that the revised manuscript will be completed within 10 days, if the experiment is restarted, it is difficult to complete the work within this time, so the experiment was not restarted.

Reviewer 3 Report

The manuscript entitled “Genetic mechanism responsible for antioxidant factors under salinity- and temperature- stress in turbot (Scophthalmus maximus)” evaluate some important antioxidant factors as superoxide dismutase (SOD), catalase (CAT) and glutathione peroxidase (GPX) on livers of Scophthalmus maximus under two important environmental factors for aquatic animals as salinity and temperature in different concentrations.

It’s a well-structured manuscript, easy to follow and understand. For improve this manuscript it’s necessary to change some part in the text and add some additional information. According to my opinion, the manuscript can be accepted for publication after minor revision. 

Author Response

Dear Reviewer:

We are very grateful for the comments from the reviewer. According to these good comments, we have revised the manuscript. The changes of the manuscript were highlighted by using “Track Changes”.

Round 2

Reviewer 1 Report

The manuscript has been performed according to previous comments. I would recommend its acceptation.

Reviewer 2 Report

The article has been significantly rephrased and restructured, and my previous objection on missing temperatures/salinities are therefore no longer significant. Now, the paper appears well structured, and I do not any longer oppose publication.